# Nucleocapsid assembly in pneumoviruses is regulated by conformational switching of the N protein

Max Renner[1], Mattia Bertinelli[1], Cédric Leyrat[1], Guido C Paesen[1], Laura Freitas Saraiva de Oliveira[1], Juha T Huiskonen[1], Jonathan M Grimes[1,2]*

[1]Division of Structural Biology, Wellcome Trust Centre for Human Genetics, University of Oxford, Oxford, United Kingdom; [2]Diamond Light Source, Didcot, United Kingdom

**Abstract** Non-segmented, (-)RNA viruses cause serious human diseases. Human metapneumovirus (HMPV), an emerging pathogen of this order of viruses (*Mononegavirales*) is one of the main causes of respiratory tract illness in children. To help elucidate the assembly mechanism of the nucleocapsid (the viral RNA genome packaged by the nucleoprotein N) we present crystallographic structures of HMPV N in its assembled RNA-bound state and in a monomeric state, bound to the polymerase cofactor P. Our structures reveal molecular details of how P inhibits the self-assembly of N and how N transitions between the RNA-free and RNA-bound conformational state. Notably, we observe a role for the C-terminal extension of N in directly preventing premature uptake of RNA by folding into the RNA-binding cleft. Our structures suggest a common mechanism of how the growth of the nucleocapsid is orchestrated, and highlight an interaction site representing an important target for antivirals.

*For correspondence: jonathan@strubi.ox.ac.uk

Competing interests: The authors declare that no competing interests exist.

## Introduction

Viruses possessing a non-segmented, single-strand, negative-sense RNA genome are the causative agents of many serious human illnesses. Notable members belonging to this group of viruses (*Mononegavirales*) include measles, rabies, Ebola, respiratory syncytial virus (RSV) and Human metapneumovirus (HMPV). HMPV (*Paramyxoviridae*, subfamily *Pneumovirinae*) is a leading cause of serious respiratory tract infections in children, the elderly, and immunocompromised individuals (*Boivin et al., 2003*; *Osterhaus and Fouchier, 2003*; *van den Hoogen et al., 2001*). In all members of the *Mononegavirales*, the RNA genome is packaged in the form of a nucleocapsid, a ribonucleo-protein complex consisting of polymerized viral nucleoproteins (N) and RNA (*Ruigrok et al., 2011*). Besides protecting the viral genome from host nucleases, the nucleocapsid serves as the template for transcription by the viral RNA-dependent RNA polymerase L. Nucleocapsid assembly necessitates a pool of monomeric, RNA-free N, termed $N^0$, which is kept in an unassembled state through an interaction with an N-terminal portion of the polymerase cofactor P, until delivered to the sites of viral RNA synthesis (*Ruigrok et al., 2011*; *Curran et al., 1995*; *Mavrakis et al., 2006*). The P protein is a multifunctional, modular protein containing large intrinsically disordered regions and is found to be tetrameric in HMPV (*Leyrat et al., 2013*). In addition, P binds to the nucleocapsid via its C–terminus, and mediates the attachment of the RNA-dependent RNA polymerase L. Furthermore in pneumoviruses, P recruits the processivity factor M2-1 (*Leyrat et al., 2014*). A great deal of effort has been spent on understanding the functions of P and recent crystal structures of P bound to N proteins ($N^0$-P) from vesicular stomatitis virus (VSV), Ebola virus, Nipah virus, and measles virus have highlighted its role in preventing assembly of N by blocking the C-terminal and N-terminal

**eLife digest** Human metapneumovirus (HMPV for short) is a major cause of infections of the airways and lungs, particularly in children, elderly individuals and people with weakened immune systems. As for all viruses, HMPV cannot survive on its own. Instead, it must invade and hijack cells in order to replicate its own genetic material and form new viruses. In HMPV, this genetic information is in the form of a strand of RNA, and is protected by a shell-like structure called a nucleocapsid. Drugs that disrupt the nucleocapsid may therefore help to kill the viruses and treat the illnesses that they cause.

Nucleocapsids are built out of many copies of a protein called nucleoprotein, which binds to a strand of RNA. However, viral nucleocapsids can only be built from nucleoproteins that are bound to viral RNA. Potentially, nucleoproteins could instead bind to RNA belonging to the cells that HMPV infects and they would then be trapped in a dead-end state. To prevent this type of unproductive binding, before the nucleocapsid is formed the nucleoprotein is kept unassembled with the help of another protein called the polymerase cofactor. However, it was not clear exactly how the polymerase cofactor helps to maintain this unassembled state.

Using techniques called cryo-electron microscopy and X-ray crystallography, Renner et al. studied the structures formed when nucleoproteins are either bound to RNA or are unassembled and bind to the polymerase cofactor. Comparing these structures revealed that RNA normally binds to a specific cleft in the nucleoprotein. However, when nucleoprotein is bound to the polymerase cofactor a portion of the nucleoprotein folds into this cleft instead, blocking the insertion of RNA. This prevents the nucleoprotein from associating with the wrong RNA, allowing the nucleoprotein to remain in an unassembled state until it is needed for the virus.

Renner et al. also found that the interactions between the nucleoprotein and the polymerase cofactor of HMPV occur at sites that are also found in several other related viruses, such as Ebola. Targeting this common region could therefore be a good strategy for developing new antiviral drugs.

extensions of N (CTD-arm and NTD-arm) which facilitate N oligomerization (*Leyrat et al., 2011*; *Guryanov et al., 2015*; *Leung et al., 2015*; *Yabukarski et al., 2014*). However, there is still paucity in our understanding of the molecular details behind the proposed mechanisms, specifically regarding how P-bound N is released, attaches to the nucleocapsid and is loaded with RNA. To address these questions in the mechanism of N-chaperoning by P and nucleocapsid assembly we performed a structural analysis of assembled and unassembled N from HMPV. Our structure of $N^0$-P reveals a conformational change, in which the negatively charged CTD-arm of N occupies the positively charged RNA binding site via specific and conserved interactions. Together with our RNA-bound structure of N these data imply a mechanism of how the growth of nucleocapsid filaments is coordinated in HMPV and related viruses.

## Results and discussion

Biochemical studies of the nucleocapsid building block N are complicated by the fact that N proteins have a strong tendency to irreversibly oligomerize and bind host nucleic acids immediately upon recombinant expression (*Gutsche et al., 2015*; *Tawar et al., 2009*). One technique to mitigate this problem is to truncate regions of N that facilitate oligomerization (*Yabukarski et al., 2014*). To stabilize monomeric full-length $N^0$ we fused the N-terminal domain of P to N, a strategy that has seen success with nucleoproteins from other viruses (*Guryanov et al., 2015*; *Kirchdoerfer et al., 2015*). We obtained crystals of RNA-free HMPV N in a monomeric state and bound to a P peptide at 1.9 Å resolution by adding trace amounts of trypsin (*Dong et al., 2007*) to prune flexible loops and promote crystallization (*Figure 1—figure supplement 1* and *Table 1*). In the structure, the P peptide is firmly nestled into a hydrophobic surface of the C-terminal domain of N (CTD) primarily composed of α-helices αC1 and αC2 (*Figure 1A,B*). Ile9, Leu10 and Phe11 of P occupy key positions and insert into this hydrophobic groove (*Figure 1B*). Unlike the $N^0$-P structure recently reported for measles (*Guryanov et al., 2015*), we find that the linker connecting N and P in our chimeric construct has

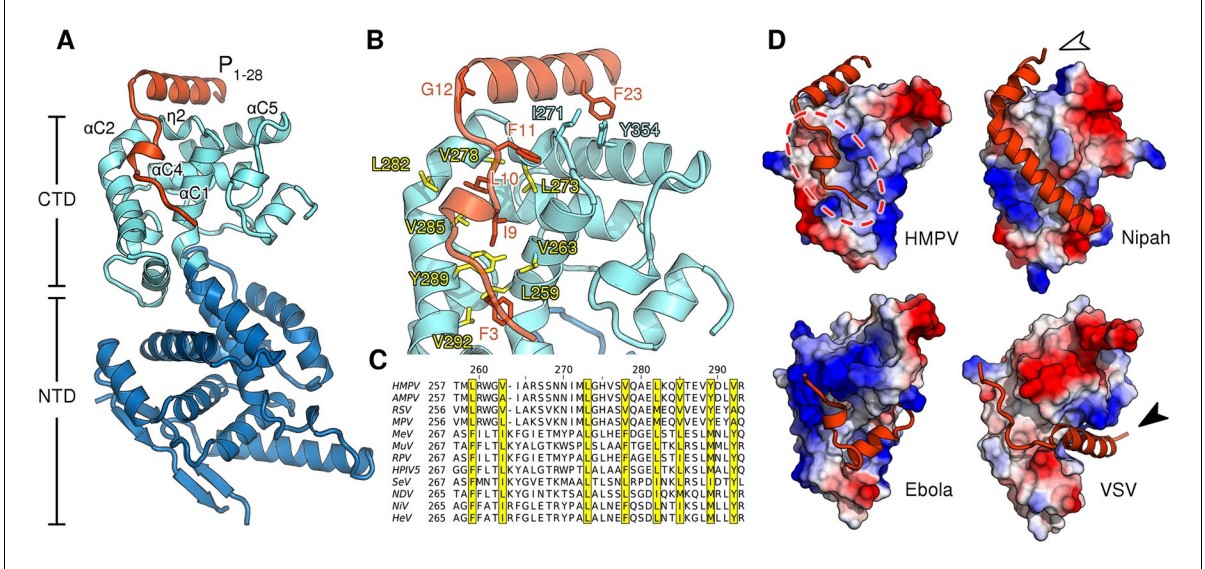

**Figure 1.** Structure of the HMPV $N^0$-P complex. (**A**) Crystal structure of RNA-free HMPV $N^0$ bound to $P_{1-28}$. The C-terminal domain (CTD) of N is colored in light blue and the N-terminal domain (NTD) in dark blue. Secondary structure elements involved in the interaction with P are indicated. The P peptide is colored in orange. (**B**) Residues that are important in facilitating the interaction between P and N are shown in stick representation. Conserved hydrophobic residues of the P binding site are colored in yellow. (**C**) Multiple sequence alignment of N proteins from *Paramyxoviridae* members. Conserved residues of the P-binding site are highlighted in yellow and correspond to those in B. Virus name abbreviations are given in Methods. (**D**) $N^0$-P complexes throughout *Mononegavirales*. Surface representations of N-CTDs of HMPV, Nipah virus (PDB ID:4CO6), Ebola virus (PDB ID:4YPI) and Vesicular stomatitis virus (PDB ID:3PMK), colored by electrostatics. CTDs are shown in the same orientation. Bound P proteins (VP35, in the case of Ebola virus) are colored in orange. The red dotted circle indicates a P-binding sub-region which is shared in all structures. Arrows are explained in the accompanying text.

The following figure supplement is available for figure 1:

**Figure supplement 1.** Construct design and purification of the HMPV $N^0$-P hybrid.

been cleaved prior to crystal growth. The P peptide wraps around the CTD and residues 12–28 form an alpha helix that lies atop N (*Figure 1B*). This helix is initiated at Gly12 and pinned to the CTD through an aromatic side-to-face interaction of Phe23 with Tyr354 of N, both residues belonging to the so-called mir motif which is conserved within *Pneumovirinae* (*Karlin and Belshaw, 2012*). This result is consistent with an earlier study, in which alanine mutations of the corresponding residues in respiratory syncytial virus resulted in a drop of polymerase activity by more than 75% in a minireplicon system (*Galloux et al., 2015*).

Alignment of *Paramyxoviridae* N sequences revealed that many hydrophobic residues lining the P-binding surface of αC1 through αC2 are shared within the family (*Figure 1C*). For all known $N^0$-P complexes (*Leyrat et al., 2011*; *Leung et al., 2015*; *Yabukarski et al., 2014*), P binds to the CTD of N (*Figure 1D*). Interestingly, although the specific interaction sites diverge (*Figure 1D*, indicated by white and black arrows), a sub-region of the CTD (*Figure 1D*, indicated by dotted circle) is bound by P in all structures, indicating that it is widely conserved throughout *Mononegavirales*.

To provide a rationale for the molecular switching between the monomeric, P-bound state and the assembled, RNA-bound state, a direct comparison at the atomic level is necessary. To this end, we purified and crystallized assembled HMPV N in the form of a decameric N-RNA ring (*Figure 2—figure supplement 1* and *Table 1*). By exploiting the ten-fold non-crystallographic symmetry in the rings, we were able to obtain excellent electron density maps at 4.2-Å resolution (*Figure 2—figure supplement 2A–C*) and build a reliable model (*Karplus and Diederichs, 2012*) (*Figure 2—figure supplement 2D*). Assembled HMPV decameric N-RNA rings are ~0.5 MDa in molecular mass and 160 Å in diameter and 70 Å in height (*Figure 2A*). The observed RNA binding mode is similar to that seen in the related RSV N-RNA structure (*Tawar et al., 2009*). The RNA wraps around the N ring and wedges tightly in the cleft between the NTD and CTD of N, which is lined by positively

**Table 1.** Data collection and refinement statistics.

| | $N^0$-P | N-RNA |
|---|---|---|
| **Data collection** | | |
| Space group | P 1 | C 2 2 $2_1$ |
| Cell dimensions | | |
| *a, b, c* (Å) | 40.9, 62.8, 86.7 | 202.0, 233.2, 203.6 |
| α, β, γ (°) | 91.0, 96.4, 109.0 | 90, 90, 90 |
| Wavelength (Å) | 0.979 | 0.917 |
| Resolution (Å) | 28.42-1.86 (1.91-1.86) | 101.19-4.17 (4.28-4.17) |
| CC (1/2) | 1.00 (0.47) | 1.00 (0.38) |
| $R_{merge}$ | 0.055 (0.590) | 0.220 (2.924) |
| I / σI | 9.2 (1.1) | 9.2 (1.0) |
| Completeness (%) | 94.8 (75.0) | 99.9 (100) |
| Redundancy | 1.7 (1.6) | 13.5 (13.8) |
| | | |
| **Refinement** | | |
| Resolution (Å) | 28.42-1.86 | 101.19-4.17 |
| No. reflections | 64451 (3743) | 36125 (2617) |
| $R_{work}$ / $R_{free}$ | 17.1/20.53 | 19.1/23.0 |
| *No. atoms* | | |
| Protein | 5707 | 27957 |
| Non-protein | 588 | 1400 |
| B-*factors* | | |
| Protein | 34.54 | 216.06 |
| Non-protein | 42.56 | 215.08 |
| *R.m.s. deviations* | | |
| Bond lengths (Å) | 0.007 | 0.010 |
| Bond angles (°) | 1.000 | 1.120 |
| *Ramachandran plot quality* | | |
| Favoured (%) | 99.72 | 95.01 |
| Allowed (%) | 0.28 | 4.96 |
| Outliers (%) | 0.00 | 0.03 |

Numbers in parentheses refer to the highest resolution shell.

$R_{free}$ was calculated as per $R_{work}$ for a 5% subset of reflections that was not used in the crystallographic refinement.

Molprobity scores are included in the Methods section.

charged residues (*Figure 2A* and *Figure 2—figure supplement 3*). In members of the *Paramyxovirinae*, the number of nucleotides in the viral genome is required to be a multiple of six (*Calain and Roux, 1993*) and the structural basis for this so-called rule of six has been elucidated recently (*Gutsche et al., 2015*). In members of the *Pneumovirinae*, however, this rule is not observed (*Tawar et al., 2009*). Our structure further highlights this difference; with each N subunit contacting seven RNA nucleotides (*Figure 2—figure supplement 2C* and *Figure 2—figure supplement 3B*).

Similar to N proteins from other members of *Mononegavirales* (*Tawar et al., 2009*; *Alayyoubi et al., 2015*; *Albertini et al., 2006*; *Green et al., 2006),* the NTD- and CTD-arms grasp the neighbouring protomers, thus facilitating assembly of polymeric N (*Figure 2B*). The NTD-arm packs against the flank of the previous protomer (*Figure 2B*, the NTD-arm of $N_{i+1}$ packs against $N_i$). The CTD-arm in turn latches onto the top of the CTD of the next protomer (*Figure 2B*, CTD-arm of $N_{i-1}$ latches onto CTD of $N_i$). We observed that the binding site of the P peptide overlaps with the

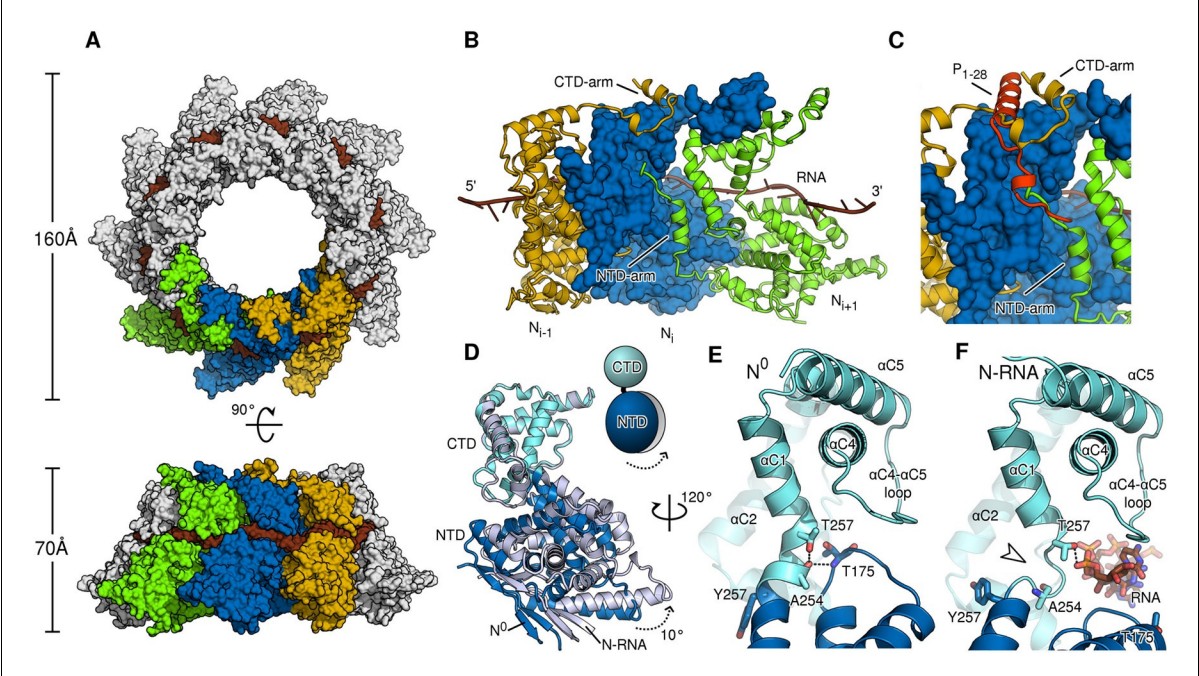

**Figure 2.** Comparison of N in assembled RNA-bound and monomeric RNA-free states. (A) top- and side-views of RNA-bound HMPV subnucleocapsid rings. N protomers and RNA are shown as surfaces with RNA rendered in brown. The diameter and height of the ring are indicated. (B) Three adjacent protomers of assembled RNA-bound N are shown viewed looking outwards from the centre of the ring, with the middle subunit rendered as surface. The exchange subdomains (NTD- and CTD-arm) that facilitate assembly of N are indicated. (C) The overlay with $P_{1-28}$ (orange) bound to the middle protomer shows that the P-binding site overlaps with that of the NTD- and CTD-arms and that binding is mutually exclusive. (D) Hinge-motion of NTD and CTD of N. Monomeric $N^0$ is superposed onto a single protomer of assembled, RNA-bound N (N-RNA, shown in grey). The NTD pivots by 10 degrees relative to the CTD (indicated). For clarity, only the NTD and CTD of the two states are shown. (E) showing $N^0$, and (F) showing N-RNA, close-up of the pivot point facilitating the hinge-motion of N. The white arrow in F indicates where the hinge region uncoils, allowing pivoting. For clarity, the P-peptide and the CTD-arm are omitted in E and F.

The following figure supplements are available for figure 2:

**Figure supplement 1.** Purification of HMPV N-RNA and characterisation of oligomeric state.
**Figure supplement 2.** Electron density maps of N-RNA.
**Figure supplement 3.** RNA-binding cleft of HMPV N.
**Figure supplement 4.** Role of a conserved aromatic residue in N hinge motion.

binding sites of the NTD- and CTD-arms (*Figure 2C*). Our structures thus provide conclusive evidence that P hampers subdomain exchange between adjacent proteins in *Pneumovirinae*. This mechanism has also been proposed for a range of viruses thoughout *Mononegavirales* (*Leyrat et al., 2011*; *Guryanov et al., 2015*; *Yabukarski et al., 2014*; *Alayyoubi et al., 2015*) and there is mounting evidence that it may be universal throughout the entire viral order.

A hinge-like motion has been proposed by which N alternates between an open, RNA-free conformation ($N^0$) and a closed RNA-bound (N-RNA) conformation (*Guryanov et al., 2015*; *Yabukarski et al., 2014*). Comparison of these two states for HMPV reveals a rigid body movement of the NTD relative to the CTD (*Figure 2D*). The conformational change rotates the NTD towards the CTD by 10°, the interface between the two domains acting as a hinge. At the interface, hinge residues Thr257 and Ala254 play a particularly crucial role. In the open, RNA-free state the hinge is maintained in a helical conformation by stabilization of Ala254 through the side chain of Thr257 and an additional backbone interaction with Thr175 (*Figure 2E*). Upon RNA binding, Thr257 contacts the backbone of a nucleotide instead of stabilizing Ala254 (*Figure 2F*). In addition, the loop containing

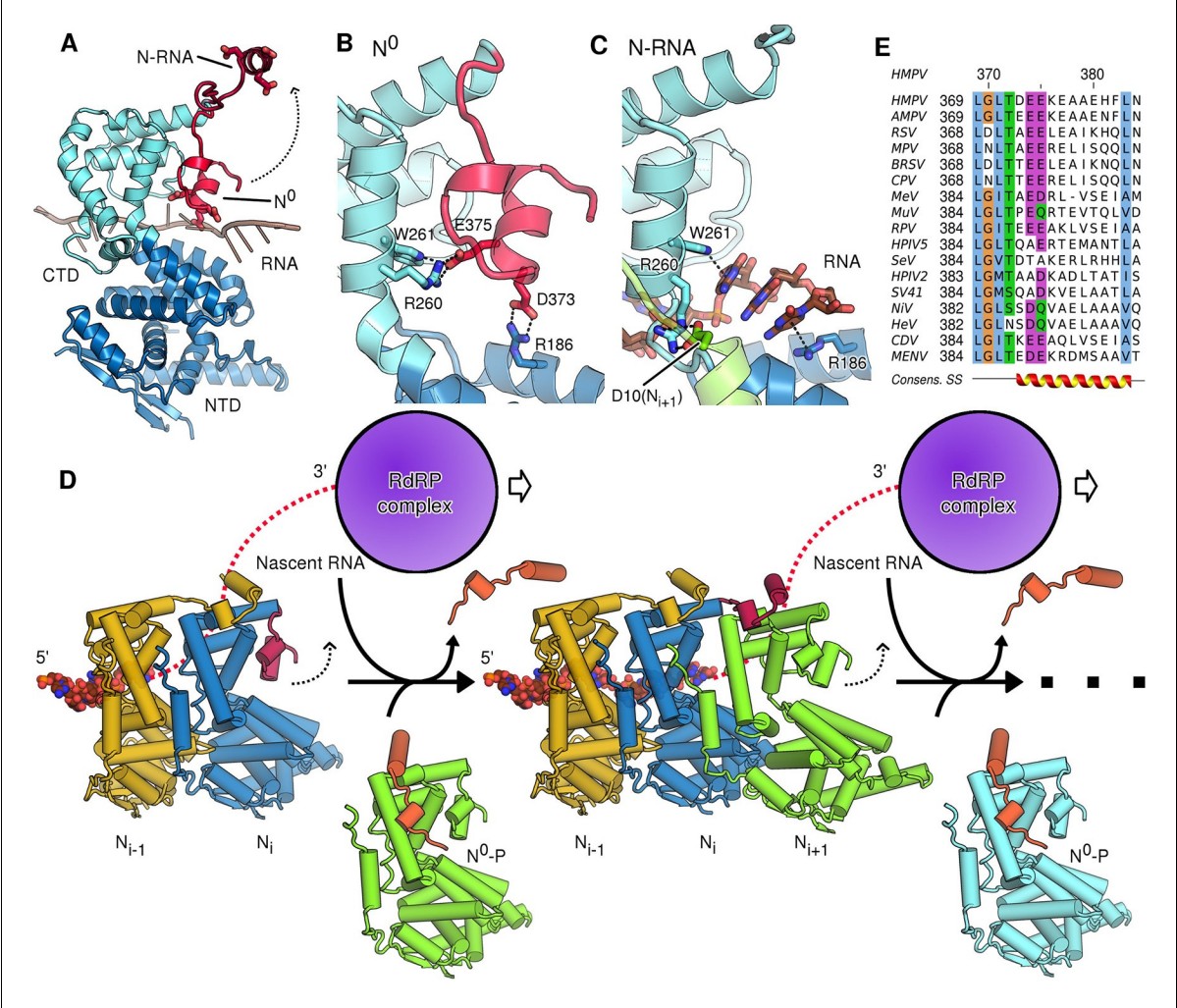

**Figure 3.** Role of the CTD-arm in inhibiting premature RNA uptake. (A) Conformational switch of the CTD-arm. The CTD-arm (red) is shown in a upward conformation assumed in the N-RNA state and downward conformation of the $N^0$ state (indicated). (B) Polar interactions fastening the CTD-arm (red) in the downward conformation. Involved residues are shown as sticks. (C) in the assembled state, the CTD-arm is displaced by RNA (shown in brown). The NTD-arm of the neighboring $N_{i+1}$ protomer is colored in green. (D) Schematic model of nucleocapsid filament growth. Nascent RNA and the active RdRP complex are indicated. Binding of emerging RNA to $N_i$ primes the displacement of P (colored in orange) and attachment of incoming $N_{i+1}$ by liberating the CTD-arm (colored in red). The dotted arrows indicate that CTD-arms switch to the upward conformation and latch onto incoming N during attachment of the next N protomer. (E) Multiple sequence alignment of CTD-arms from *Paramyxoviridae* family members. Residues are colored using the ClustalX color scheme. The consensus secondary structure is indicated below the alignment. Virus name abbreviations are given in Materials and methods.

The following figure supplement is available for figure 3:

**Figure supplement 1.** CTD-arms in other *Mononegavirales* family members.

Thr157 retracts to sterically accommodate the RNA chain. Having lost the stabilizing contacts of Thr257 and Thr157, the helical hinge region around Ala254 unravels and becomes flexible (*Figure 2F*, indicated by white arrow), allowing the relative domain motions of NTD and CTD. Furthermore, we propose that Tyr252 is important in facilitating the hinge motion. Tyr252 is positioned just before the pivot point and packs tightly against αC3 (*Figure 2—figure supplement 4A*). An aromatic residue at this position is found packing against the same helix in most known structures of N (*Figure 2—figure supplement 4B–G*). Transition from the RNA-free to RNA-bound state induces a rotation of αC3, exerting upwards pressure on Tyr252 that is conferred onto the NTD (*Figure 2—*

*figure supplement 4A*). Intriguingly, in structures of *Paramyxovirinae* N, which obey the rule-of-six, this aromatic is flipped in the opposite direction (*Figure 2—figure supplement 4H,I*) and contacts RNA (*Gutsche et al., 2015*), suggesting a similar coupling of RNA-binding and hinge-motion in these viruses.

The most profound changes between assembled and unassembled states, however, involve the CTD-arm of N, a region that has been little characterized in pneumoviruses. In the polymeric, RNA-bound state of N (N-RNA) the CTD-arm flips upwards and latches onto the next protomer, whilst in the monomeric state ($N^0$) it packs down against the core of N (*Figure 3A*). The downward, monomeric conformation is stabilized by specific salt-bridges linking the CTD-arm with the core of N (*Figure 3B*). In this position the negatively charged CTD-arm folds into the positively charged RNA binding cleft, occupying it and directly blocking the binding of RNA (*Figure 3A*). It is interesting to note, that whilst the CTD-arm blocks the RNA site in HMPV, it is the P peptide that inserts itself there in VSV (*Leyrat et al., 2011*). Because this is not observed in paramyxoviral $N^0$-P complexes (*Guryanov et al., 2015*; *Yabukarski et al., 2014*) we hypothesize that, in *Rhabdoviridae*, a different strategy has evolved to block off the RNA binding cleft. The question arises how the interactions that hold the downwards-positioned CTD-arm in place are broken when assembly of N-RNA necessitates it flipping into the upwards position. In the RNA-free state, Arg260 and Trp261 contact Glu375, while Arg186 forms a salt-bridge with Asp373 of the CTD-arm (*Figure 3B*). In the assembled, RNA-bound state these interactions are broken, with Arg186 and Trp261 now positioning RNA nucleotides in the cleft, whilst Arg260 instead fastens onto the NTD-arm of the neighbouring $N_{i+1}$ (*Figure 3C*). The shift from initial stabilization of the inhibitory (downwards) CTD-arm conformation to stabilization of bound RNA and neighbouring N subunit implies that attachment of a new N protomer and insertion of nascent RNA occur concomitantly. This makes sense in the context of viral replication sites, where tetrameric P proteins act as molecular chaperones attaching to the nucleocapsid template, polymerase and free $N^0$, leading to high local concentrations of nucleoprotein and RNA.

Based on the comparison of our N-RNA and $N^0$-P structures we suggest a model for nucleocapsid growth (*Figure 3D*). Upon delivery of fresh $N^0$-P to the growth site, addition of the next N protomer ($N_{i+1}$) to the filament necessitates that the CTD-arm of the terminal $N_i$ unbinds and flips upwards (*Figure 3D*, indicated by dotted arrow), latching onto $N_{i+1}$ and displacing P. In our model, this is driven by the formation of new interactions to the NTD- and CTD arms and, importantly, the concerted insertion of nascent RNA into the RNA binding cleft of $N_i$, with the CTD-arm switching into the upward conformation. In this model the growth of the filament is reminiscent of a zipper closing up with one row of teeth corresponding to nascent viral RNA and the other to newly delivered N subunits which interdigitate in a fluid, concerted motion. The notion that concerted RNA insertion is required for the hand-over of N subunits from P lends additional specificity to the nucleocapsid polymerization reaction.

We hypothesized that the role of the CTD-arm in inhibiting premature RNA binding may be conserved and therefore compared sequences throughout *Paramyxoviridae* (*Figure 3E*). We find a semi-conserved LGLT-motif within the CTD-arms which is followed by a stretch of residues with helical propensity. The beginning of this stretch preferentially features negatively charged residues at positions equivalent to HMPV which may in turn pack against the complementary charges of the RNA binding cleft. Indeed, analysis of structures of more distantly related members of *Mononegavirales* shows that these negatively charged residues are topologically conserved and that a switch to the downward conformation would position these residues into the RNA binding cleft (*Figure 3—figure supplement 1*).

In conclusion, the reported structures of a paramyxoviral N protein reveal two distinct conformational states, N bound either to the polymerase cofactor P or to RNA. A direct comparison of these two structures provides a molecular level rationale for how nucleocapsid assembly is controlled through P by sterically blocking the binding sites of the NTD- and CTD-arms. In addition, this work elucidates a key role of the CTD-arm in hindering premature RNA insertion into the binding cleft, thus presenting a mechanistic explanation of how premature RNA uptake is directly inhibited in *Paramyxoviridae*. Peptides of the $N^0$-binding region of P have previously been shown to inhibit replication activity in RSV (*Galloux et al., 2015*), Nipah virus (*Yabukarski et al., 2014*), and rabies virus (*Castel et al., 2009*). The characterization of P-binding surfaces on N proteins is therefore of biomedical importance as these surfaces constitute genuine targets for the development of antivirals.

## Materials and methods

### Expression and purification of N-RNA rings

The full-length N gene from human metapneumovirus (strain NL1-00, A1) was cloned into the pOPINE expression vector, which includes a C-terminal His-tag, using the In-Fusion system (Takara Clontech, Mountain View, CA) following standard procedures. The construct was verified by sequencing. Rosetta2 E.coli cells harboring the expression plasmid were grown at 37°C in terrific broth containing appropriate antibiotics and expression was induced at an OD600 of 0.8 by adding isopropyl β-D-1-thiogalactopyranoside to 1 mM. The temperature was then lowered to 18°C and after further 18 hrs the cells were harvested by centrifugation (18°C, 20 min, 4000 x g). Cell pellets were resuspended in 40 mL of 25 mM Tris, pH 8, 1 M NaCl per L of culture and lysed by sonication. The lysate was centrifuged (4°C, 45 min, 50000 x g) and the supernatant was filtered and loaded on a column containing pre-equilibrated $Ni^{2+}$-nitrilotriacetic (NTA) agarose (Qiagen, Netherlands). The column was washed and the protein was eluted in 25 mM Tris, pH 8, 1 M NaCl, 400 mM imidazole. The eluate was further purified by size exclusion chromatography using a Superose6 10/300 column (GE Healthcare, United Kingdom) equilibrated in 25 mM Tris, pH 8, 1 M NaCl. The protein was buffer exchanged into 25 mM Tris, pH 8, 150 mM NaCl, 500 mM NDSB201, 50 mM Arginine using a PD10 column (GE Healthcare) and then concentrated to ~4 mg/mL for crystallization.

### Expression and purification of the N0-P hybrid

The $N^0$-P hybrid gene was generated by fusing the sequence corresponding to the first 40 residues of HMPV P (strain NL1-00, A1) to the 3' end of the full-length N gene using overlapping primer PCR. The resulting hybrid construct was cloned into POPINE as described above and verified by sequencing. Protein expression was carried out as described for N, above. Cell pellets were resuspended in 20 mM Tris, pH 7, 1M NaCl, lysed by sonication and the lysate was subsequently centrifuged (4°C, 45 min, 50000 x g). The supernatant was purified using a column containing pre-equilibrated $Ni^{2+}$-NTA agarose and elution was carried out using 20 mM Tris, pH 7, 1M NaCl, 300 mM imidazole. The protein was then buffer exchanged into 20 mM Tris, pH 7, 100 mM NaCl and loaded onto a HiTrap Heparin HP column (GE Healthcare) for further purification using a stepwise NaCl gradient. Finally, the N0-P hybrid was gel-filtrated using a Superdex 75 column (GE Healthcare) equilibrated with 20 mM Tris, pH 7, 100 mM NaCl, and concentrated to ~7 mg/mL for crystallization.

### Crystallization and data collection

Sitting drop, vapor diffusion crystallization trials were set up in 96-well Greiner plates using a Cartesian Technologies robot (*Walter et al., 2005*). A diamond-like, diffraction quality N-RNA crystal was obtained after 132 days in mother liquor containing 100 mM Tris/Bicine, pH 8.5, 90 mM NPS ($NaNO_3$, $Na_2HPO_4$, $(NH_4)_2SO_4$), 37.5% methyl-2 4-pentanediol, polyethylene glycol 1000 and polyethylene glycol 3350 of the MORPHEUS crystal screen. The crystal was frozen in liquid nitrogen and diffraction data up to 4.2 Å were recorded at 100 K on the I04-1 beamline at Diamond Light Source, Didcot, UK.

For the N0-P hybrid, crystals were obtained via in-situ proteolysis (*Dong et al., 2007*) using 1 μg of trypsin per 1000 μg of sample. The trypsin was added to the concentrated $N^0$-P preparation just before setting up the crystallization trials. Initial crystals formed in mother liquor containing 100 mM PCB System, pH 7, 25% polyethylene glycol 1500 and improved crystals could be grown with additives of the Hampton Silver Bullet screen (9 mM 1,2-diaminocyclohexane sulfate, 6 mM diloxanide furoate, 17 mM fumaric acid, 10 mM spermine, 9 mM sulfaguanidine and 20 mM HEPES, pH 6.8). The crystals were cryoprotected in 25% glycerol and frozen in liquid nitrogen. Diffraction data up to 1.9 Å were recorded at 100 K on the I04 beamline at Diamond Light Source, Didcot, UK. All data were processed and scaled with XIA2 (*Winter, 2010*).

### Structure determination and refinement

The structure of $N^0$-P was solved by molecular replacement using PHASER (*McCoy et al., 2007*) with the structure of RSV N (*Tawar et al., 2009*) as a search model. Iterative rounds of refinement using PHENIX (*Adams et al., 2010*) with TLS parameters and manual building in COOT (*Emsley and Cowtan, 2004*) resulted in a model for HMPV N starting at residue 30 and ending at residue 383 of

the total 394. Residues 101 to 111 were found to be disordered and were not included in the model. Of the 40 P residues contained in our $N^0$-P construct the first 28 were well-resolved.

The structure of the RNA-bound subnucleocapsid ring was solved with PHASER (*McCoy et al., 2007*) using a decameric model of our high-resolution HMPV N structure as a search model. Initially, we performed iterative rounds of manual building with COOT (*Emsley and Cowtan, 2004*) and refinement using PHENIX (*Adams et al., 2010*) with non-crystallographic symmetry (NCS) constraints to lower the parameter to observations ratio. To aid model building we made use of density modified maps obtained with PHENIX RESOLVE (*Adams et al., 2010*) and Parrot of the CCP4 suite (*Winn et al., 2011*) in combination with B-factor sharpening. Later stages of refinement were performed with autoBuster (*Smart et al., 2012*), applying NCS restraints, TLS parameters and using our high-resolution $N^0$-P structure to generate reference model restraints. Structures were validated with MolProbity (*Chen et al., 2010*) resulting in overall MolProbity scores of 0.95 and 2.22 for $N^0$-P (at 1.9 Å) and N-RNA (at 4.2 Å), respectively. Refinement and geometry statistics are given in *Table 1*.

## Multiple sequence alignment

Multiple sequence alignments (MSA) were carried out with PROMALS3D (*Pei and Grishin, 2014*) and figures were prepared with Jalview. Nucleoprotein sequences of the following viruses were used: HMPV, Human metapneumovirus, AMPV, Avian metapneumovirus, RSV, Respiratory syncytial virus, MPV, Murine pneumonia virus, BRSV, Bovine respiratory syncytial virus, CPV, Canine pneumonia virus, MeV, Measles virus, MuV, Mumps virus, RPV, Rinderpest virus, HPIV5, Human parainfluenza virus 5, SeV, Sendai virus, HPIV2, Human parainfluenza virus 2, SV41, Simian virus 41, NiV, Nipah virus, HeV, Hendra virus, CDV, Canine distemper virus, MENV, Menangle virus.

## Electron microscopy

N-RNA rings were analysed via electron cryomicroscopy (cryo-EM). Aliquots (3 µl) of N-RNA preparations were pipetted onto glow-discharged Cflat holey carbon grids (Protochips, Raleigh, NC) and excess liquid was blotted with filter paper for 3 s. Grids were then plunge-frozen in an ethane-propane mixture at liquid nitrogen temperature using a CP3 plunging device (Gatan). Cryo-EM data were acquired using a 300-kV Polara transmission electron microscope (FEI) equipped with a K2 Summit direct electron detector (Gatan) and using defocus values ranging from -2.0 to -6.0 µm at a calibrated magnification of 37,000x, resulting in a pixel size of 1.35 Å. The contrast transfer function (CTF) parameters were determined using CTFFIND3 (*Mindell and Grigorieff, 2003*) and 2D-classification was carried out with RELION (*Scheres, 2012*).

## Acknowledgements

We thank Dr Ron Fouchier and Dr Bernadette van den Hoogen for providing us with the plasmids encoding HMPV N and P, Dr Alistair Siebert for help with electron microscopy, Dr Luigi De Colibus, Dr Jonathan Elegheert, Dr Kamel El Omari, Prof David Stuart for helpful discussions and Sina Wittmann for proofreading the manuscript. The presented research has received funding from the European Union Seventh Framework Programme (FP7/2007-2013) under SILVER grant agreement No 260644. Administrative support came from the Wellcome Trust Core award (090532/Z/09/Z), MR was supported by a Wellcome Trust Studentship (099667/Z/12/Z), and JTH by the Academy of Finland (130750 and 218080) and by the European Research Council (ERC) under the European Union's Horizon 2020 research and innovation programme (grant agreement No 649053). MB was funded by Instruct, part of the European Strategy Forum on Research Infrastructure (ESRFI). The OPIC electron microscopy facility was founded by a Wellcome Trust JIF award (060208/Z/00/Z) and is supported by a WT equipment grant (093305/Z/10/Z). We thank Diamond Light Source for beamtime (proposal MX8423) and the staff of beamlines I04 and I04-1 for assistance with crystal testing and data collection. The coordinates and structure factors for HMPV $N^0$-P and N-RNA have been deposited in the Protein Data Bank (PDB) under the accession codes 5fvd and 5fvc, respectively. The authors declare no competing financial interests.

## Additional information

### Funding

| Funder | Grant reference number | Author |
|---|---|---|
| Wellcome Trust | 090532/Z/09/Z | Max Renner<br>Mattia Bertinelli<br>Cedric Leyrat<br>Guido C Paesen<br>Laura Freitas Saraiva de Oliveira<br>Juha T Huiskonen<br>Jonathan M Grimes |
| Wellcome Trust | 099667/Z/12/Z | Max Renner |
| European Commission | 260644 | Max Renner<br>Mattia Bertinelli<br>Cedric Leyrat<br>Guido C Paesen<br>Laura Freitas Saraiva de Oliveira<br>Juha T Huiskonen<br>Jonathan M Grimes |
| Suomen Akatemia | 130750 | Juha T Huiskonen |
| Wellcome Trust | 060208/Z/00/Z | Max Renner<br>Mattia Bertinelli<br>Cedric Leyrat<br>Guido C Paesen<br>Laura Freitas Saraiva de Oliveira<br>Juha T Huiskonen<br>Jonathan M Grimes |
| Wellcome Trust | 093305/Z/10/Z | Max Renner<br>Mattia Bertinelli<br>Cedric Leyrat<br>Guido C Paesen<br>Laura Freitas Saraiva de Oliveira<br>Juha T Huiskonen<br>Jonathan M Grimes |

The funders had no role in study design, data collection and interpretation, or the decision to submit the work for publication.

### Author contributions

MR, MB, CL, JTH, JMG, Conception and design, Acquisition of data, Analysis and interpretation of data, Drafting or revising the article; GCP, Acquisition of data, Drafting or revising the article; LFSdO, Acquisition of data, Analysis and interpretation of data, Drafting or revising the article

### Author ORCIDs

Jonathan M Grimes, http://orcid.org/0000-0001-9698-0389

# Additional files

### Major datasets

The following datasets were generated:

| Author(s) | Year | Dataset title | Dataset URL | Database, license, and accessibility information |
|---|---|---|---|---|
| Renner M, Bertinelli M, Leyrat C, Paesen GC, Saraiva de Oliveira LF, Huiskonen JT, Grimes JM | 2016 | Structure of RNA-bound decameric HMPV nucleoprotein | https://www.ebi.ac.uk/pdbe/ | 5fvc |

Renner M, Bertinelli M, Leyrat C, Paesen GC, Saraiva de Oliveira LF, Huiskonen JT, Grimes JM 2016 Human metapneumovirus N0-P complex https://www.ebi.ac.uk/pdbe/ 5fvd

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
