## [Decision Letter]

[Editors’ note: this article was originally rejected after discussions between the reviewers, but the manuscript was accepted after an appeal against the decision.]

Thank you for submitting your work entitled "Nucleocapsid assembly in pneumoviruses is regulated by conformational switching of the N protein" for consideration by *eLife*. Your article has been reviewed by two peer reviewers, and the evaluation has been overseen by a Reviewing Editor and John Kuriyan as the Senior Editor. Our decision has been reached after consultation between the reviewers and the Reviewing Editor. Based on these discussions, we regret to inform you that your work will not be considered further for publication in *eLife*. The decision was made not because of the quality of the work, which both reviewers judged to be fine, but rather because of the criteria for novelty and breadth of interest that the journal and its Board of Reviewing Editors have set for the journal.

Renner and colleagues provide evidence that nucleocapsid assembly in pneumoviruses involves a conformational switch of the N protein. That conclusion is based on the structures of an N-protein RNA complex and a soluble form of N, referred to as No, both for human metapneumovirus (HMPV) N. The No structure was obtained in complex with a small N-terminal fragment of P. The structures are of interest and suggest that a conformational change occurs during the process of encapsidation of the nascent RNA. The authors propose, based on the structures, that the C-terminal arm of the last N protein molecule in the growing filament flips up to allow engagement of the new incoming No-P, displacing P concomitant with the insertion of the nascent RNA chain into the newly added N. The process of encapsidation of the nascent RNA lacks a detailed molecular model, and this manuscript may provide some important new clues as to how the process occurs not just for HMPV, but perhaps other viruses in the order mononegavirales (MNV). There are, however, some concerns regarding the No-P structure, and several statements of novelty and precedent in the manuscript that do not seem justified. Moreover, other structures of RNA free, soluble No and N-RNA complexes have been determined for other viruses in the MNV, diminishing the novelty of present work.

1) The No-P structure was solved using an artificial construct in which the N-terminal 40 residues of P were fused to the C-terminal arm of full-length N, and crystals obtained by in situ proteolysis with trypsin. It is uncertain to what extent the structure obtained represents a physiologically relevant form of soluble N or is in fact a form that is simply obtained using the in situ proteolytic approach used here.

2) The conformational switch model proposed suggests contacts that can be tested experimentally using HMPV reverse genetics – this would go a long way to alleviate the concern of point 1.

3) Throughout the manuscript the authors make broad claims of precedent, stating that this is the first virus in the mononegavirales for which structures of both No and N-RNA have been determined. This is not the case – the authors show several previously published structures in Figure 1. Such statements of precedent should be removed or clarified (see Abstract for example).

---

## [Author Response]

*Renner and colleagues provide evidence that nucleocapsid assembly in pneumoviruses involves a conformational switch of the N protein. That conclusion is based on the structures of an N-protein RNA complex and a soluble form of N, referred to as No, both for human metapneumovirus (HMPV) N. The No structure was obtained in complex with a small N-terminal fragment of P. The structures are of interest and suggest that a conformational change occurs during the process of encapsidation of the nascent RNA. The authors propose, based on the structures, that the C-terminal arm of the last N protein molecule in the growing filament flips up to allow engagement of the new incoming No-P, displacing P concomitant with the insertion of the nascent RNA chain into the newly added N. The process of encapsidation of the nascent RNA lacks a detailed molecular model, and this manuscript may provide some important new clues as to how the process occurs not just for HMPV, but perhaps other viruses in the order mononegavirales (MNV). There are, however, some concerns regarding the No-P structure, and several statements of novelty and precedent in the manuscript that do not seem justified. Moreover, other structures of RNA free, soluble No and N-RNA complexes have been determined for other viruses in the MNV, diminishing the novelty of present work.*

I would like to thank you and the referees for the comments regarding our manuscript “Nucleocapsid assembly in pneumoviruses is regulated by conformational switching of the N protein”. Two of the three main points brought forward by the referees are either unjustified or easy to address.

*1) The No-P structure was solved using an artificial construct in which the N-terminal 40 residues of P were fused to the C-terminal arm of full-length N, and crystals obtained by in situ proteolysis with trypsin. It is uncertain to what extent the structure obtained represents a physiologically relevant form of soluble N or is in fact a form that is simply obtained using the in situ proteolytic approach used here.*

Linking of binding partners as we have performed in this manuscript is a powerful technique to circumvent problems of unstable binding and increase naturally occurring interaction. The method (reviewed in Sivaraman et al., 2013) is well-established and has been successfully used on a variety of systems, including MHC-receptors and, indeed, viral nucleoproteins (e.g. Kirchdoerfer et al., 2015). Due to the inherent propensity to form oligomeric assemblies such tricks are absolutely necessary for the study of N. The referees categorically denote the method as being artificial but offer no specific criticisms to the structure at hand. Similarly, the use of limited proteolytic digestion is a very common technique in crystallography to facilitate the pruning of flexible loops and thus promote crystallization (Dong et al., 2007). It seems that the major criticism of the referees was that we utilized two common and established techniques which happen to be unfamiliar to them. Indeed, we can bring forward several points that confirm our N0-P structure is correctly folded, even though this should not be necessary: ( (1)) the binding site of P is in line with previous functional studies (Galloux et al., 2015), as mentioned in the manuscript, ( (2)) the P-site is also in line with structures of related viruses, ( (3)) the linker between N and P is cleaved by the in-situ proteolysis yielding non-covalently bound P which should completely negate this point of contention, ( (4)) the overall fold of N follows other N proteins. We therefore cannot understand why our structure is deemed artificial and would like to challenge the referees to bring forward specific criticisms. We addressed the criticisms in our manuscript and explain the rationale of using a chimeric protein and proteolytic digestion. We have also addressed the criticism of data/model quality for our N-RNA structure by including CC*, CCwork and CCfree statistics by resolution shell (Karplus and Diederichs, 2012).

*2) The conformational switch model proposed suggests contacts that can be tested experimentally using HMPV reverse genetics* – *this would go a long way to alleviate the concern of point 1.*

We fully agree that testing the proposed interactions using a reverse genetics system is key in further dissecting the mechanism of nucleocapsid assembly. Unfortunately, we lack the resources or necessary collaborations to perform these experiments. This is why our manuscript was written as a short report focused on biophysical and structural analysis of this system and we feel that such experiments would be beyond its scope. However, we could conceive of mutagenesis experiments that could test the interactions in-vitro, and these may provide valuable additional validation

*3) Throughout the manuscript the authors make broad claims of precedent, stating that this is the first virus in the mononegavirales for which structures of both No and N-RNA have been determined. This is not the case* – *the authors show several previously published structures in Figure 1. Such statements of precedent should be removed or clarified (see Abstract for example).*

Our study aims at structurally characterising assembly of N. The transition between assembled N-RNA and monomeric N0 -P was analysed in-depth in our manuscript. At the point of review, to our best knowledge, there was no instance where an assembled N-RNA and monomeric N0 -P structure was available for the same virus. The only instance where this might be contended is for vesicular stomatitis virus (VSV). However, in this case both structures form oligomeric, assembled rings that are conformationally identical (Leyrat et al., 2011). We could thus, for the first time, analyse the structural transition without having to resort to qualitative comparisons with phylogenetically distant homologues. Similar to major comment #1 the referees unfairly dismissed our claims without specific criticism. We would like to challenge them to show us for which virus they think monomeric N0 -P and assembled N-RNA was available during review. However, during to the lengthy review process, the measles virus N0 -P was published (Guryanov et al., 2015, published online on December 30), meaning that we cannot make the contended claim anymore. To quote the manuscript by Guryanov et al: "Here, for the first time in Paramyxovirus research, our data allow direct comparison of the structure of the nucleoprotein from the same virus in two functional states: a P-bound naive state, and an RNA-bound helical assembly." Consequently, although they were true during the review stage, we have now removed these claims of precedence.